# β-Phenylethylamine as a Natural Food Additive Shows Antimicrobial Activity against *Listeria monocytogenes* on Ready-to-Eat Foods

**DOI:** 10.3390/foods9101363

**Published:** 2020-09-25

**Authors:** Francis Muchaamba, Roger Stephan, Taurai Tasara

**Affiliations:** Institute for Food Safety and Hygiene, Vetsuisse Faculty, University of Zürich, 8057 Zürich, Switzerland; stephanr@fsafety.uzh.ch (R.S.); tasarat@fsafety.uzh.ch (T.T.)

**Keywords:** β-phenylethylamine, antimicrobial, *Listeria monocytogenes*, bologna type sausage, cold smoked salmon

## Abstract

*Listeria monocytogenes* is an important foodborne pathogen and a major cause of death associated with bacterial foodborne infections. Control of *L. monocytogenes* on most ready-to-eat (RTE) foods remains a challenge. The potential use of β-phenylethylamine (PEA) as an organic antimicrobial against *L. monocytogenes* was evaluated in an effort to develop a new intervention for its control. Using a collection of 62 clinical and food-related isolates we determined the minimum inhibitory concentration (MIC) of PEA against *L. monocytogenes* in different broth and agar media. Bologna type sausage (lyoner) and smoked salmon were used as food model systems to validate the in vitro findings. PEA had a growth inhibitory and bactericidal effect against *L. monocytogenes* both in in vitro experiments as well as on lyoner and smoked salmon. The MIC’s ranged from 8 to 12.5 mg/mL. Furthermore, PEA also inhibited *L. monocytogenes* biofilm formation. Based on good manufacturing practices as a prerequisite, the application of PEA to RTE products might be an additional hurdle to limit *L. monocytogenes* growth thereby increasing food safety.

## 1. Introduction

*Listeria monocytogenes* is an important foodborne pathogen and a major cause of death associated with bacterial foodborne infection [1,2]. An association between contaminated ready-to-eat (RTE) food in outbreaks and sporadic listeriosis infections is well documented [1,2,3,4,5,6,7,8,9]. Deli meats are amongst the most common foods linked to such outbreaks, an example being the recent and largest listeriosis outbreak in history recorded in South Africa, where polony, a commonly consumed bologna type sausage deli meat product, was implicated as the source of the outbreak [3,8,10,11,12,13,14]. Recent data from the European Union (EU) member states show that *L. monocytogenes* is isolated at different frequencies from RTE products [2,15]. The occurrence of *L. monocytogenes* in RTE foods coupled with the increase of susceptible populations and changing eating patterns that favor high risk foods such as RTE foods like salads and deli meats raises concerns [16,17,18,19]. As a result, a zero-tolerance policy for *L. monocytogenes* on RTE foods has been adopted by countries such as the USA [20,21]. Meanwhile, the EU has adopted a stern policy concerning *L. monocytogenes* in foodstuffs which can support the growth of this bacterium or are intended for the high-risk populations [22]. This has forced food processing companies to employ a variety of hurdle procedures in an effort to ensure that their food products are *L. monocytogenes*-free [23,24,25]. Several technologies and intervention strategies aimed at inhibiting or killing these pathogens thereby improving the microbiological quality have been employed albeit with varying levels of success [26,27,28,29,30,31,32,33,34,35,36].

High hydrostatic pressure (HHP), phage biocontrol, bacteriocin-producers, or bioprotective cultures, nisin, sodium chloride, sodium lactate, and other food-grade antimicrobials are some of the treatments and techniques used to increase the safety of RTE food products [28,29,30,31,32,33,37,38,39,40]. These measures can prolong product shelf life, which is critical due to the globalization of the food industry, as food is often transported over long distances before consumption [41]. However, with prolonged use, bacteria tend to become resistant to most such interventions or antimicrobial treatments. This is well documented for interventions such as salt and nisin to mention a few [42,43,44,45,46,47,48]. In the case of acid, cold and salt resistance, bacteria pre-exposed to these stresses can be inadvertently primed for gastric stress and intracellular life increasing their virulence e.g., they survive the acid barrier of the stomach much better [49,50,51,52]. Moreover, most of these treatments have the limitation that many consumers are favoring products with less of these treatments, while producers are aspiring to limit the use of additives, especially nonorganic ones [29].

*L. monocytogenes* control is further compounded by the bacterium’s ability to survive harsh conditions commonly encountered in foods which include low temperature, low water activity, oxidative and osmotic stress, low pH, or desiccation [37,53,54,55,56]. In addition, this bacterium can grow at refrigeration temperature, which is particularly significant in cases of temperature abuse [7,24,25,42,48,57,58]. To achieve this *L. monocytogenes* deploys various regulatory protein systems that control the implementation of the necessary stress adaption and virulence responses, which allow stress survival and transmission along the food chain as well as subsequently target host infection and pathogenicity [46,54,55,56]. Moreover, the common occurrence of *L. monocytogenes* in the environment makes its introduction into processing plants highly likely. Once introduced, several factors including biofilm production and tolerance of disinfectants increase the probability of *L. monocytogenes* to establish and persist in a facility [3,14,54,59,60,61,62]. As such, *L. monocytogenes* continues to be a costly control challenge for the food industry especially given its ability to persist in food producing facilities for long periods, in some cases, eventually causing outbreaks [14,35,60,61,62,63,64].

In an aim to mitigate the impacts of food contamination, manufacturers and researchers are therefore exploring new avenues to improve food safety and eliminate *L. monocytogenes* from their facilities. Current research on the development of novel techniques to reduce bacterial contamination of food is aimed at substances or interventions that can alter growth phenotypes of bacteria increasing shelf life and safety of foods [29,65]. Moreover, the search is being directed towards natural substances as consumers are becoming more particular of what their food contains with an increased demand for more organic natural food on the rise in the market [33,38,66,67]. One such substance that fits in this natural additive category is β-phenylethylamine (PEA) that has been shown to have antimicrobial activity against bacteria such as *Escherichia coli, Pseudomonas aeruginosa,* and *Staphylococcus aureus* in settings such as beef, biofilms, and clinical environments (as a liquid catheter flush) [65,68,69]. Though PEA is at present not accepted as a food preservative in the EU, it would be ideal for food applications due to its GRAS status (Generally Regarded As Safe) and does not constitute a known health hazard to humans [65,68,69]. PEA is, however, currently approved for use as a flavoring agent in several countries including those of the EU [70,71]. Moreover, PEA is a relatively inexpensive naturally occurring trace substance found in chocolate [65,69], where its concentration increases during cocoa fermentation [65,72]. PEA is currently sold by many health companies for various reasons including mood enhancement and to aid weight loss, with its intake as a nutritional supplement currently recommended at 500 mg per day or more in some cases [68,69].

We recently observed inhibition patterns of *L. monocytogenes* upon PEA inclusion to growth media under alkaline stress conditions [73,74]. This suggested that PEA could also be potentially exploited in developing novel ways to control *L. monocytogenes* in foods. In this study, we took advantage of this observation and used it to establish the minimum inhibitory and bactericidal concentration of PEA against *L. monocytogenes*. Furthermore, the practical applicability of this concept was further tested on cold smoked salmon and a bologna type sausage product (lyoner), for the development of a novel control strategy against *L. monocytogenes* on such RTE products. This was motivated by the fact that a type of bologna sausage called polony and other processed meats were recently associated with the biggest listeriosis outbreak in history, with more than 1000 confirmed cases, and at least 216 deaths [10]. Furthermore, seafood makes up a significant portion of food recalled due to *L. monocytogenes* contamination [2,15]. Control and prevention of *L. monocytogenes* contamination of cold-smoked fish products such as cold-smoked salmon continues to be a challenge mainly due to the intrinsic characteristics of these products that support the growth of *L. monocytogenes* [39,75,76,77,78]. Therefore, the development of novel control strategies for *L. monocytogenes* in bologna type sausages and cold-smoked salmon is of crucial importance for these industries.

## 2. Materials and Methods

### 2.1. Bacterial Strains and Culture Conditions

Sixty-two strains covering the biodiversity of *L. monocytogenes* including food-derived isolates, outbreak and sporadic listeriosis isolates as well as EGDe the *L. monocytogenes* reference strain were analyzed (Table 1). The study strains used covered clinical and food relevant genetic backgrounds of *L. monocytogenes* (Appendix A). Frozen stocks of the strains were kept in a cryo-preservative mixture of brain heart infusion medium (BHI, Oxoid, UK) and glycerol (20%) at −80 °C. Aliquots from the frozen stock were streaked out on BHI or blood agar plates and incubated overnight at 37 °C to get single colonies. To prepare working inoculum, single colonies of each strain were picked and pre-cultured twice (16 h, 37 °C and 150 rpm) in BHI broth, generating secondary stationary phase cultures that were subsequently used in our experiments.

### 2.2. Growth Evaluation and MIC Determination

Secondary cultures prepared from each strain were diluted in BHI to 10^7^ colony forming units (CFU)/mL. To evaluate growth under β-Phenylethylamine hydrochloride (PEA), (Sigma-Aldrich Chemie GmbH, Steinheim, Germany) and determine the minimum inhibitory concentration (MIC), 100 μL of BHI containing 0, 3.13, 6.25, 12.5, 16, 20, 25, 50, 100, and 150 mg/mL PEA were pipetted in triplicate to a 96-well microplate that had been pre-filled with 100 μL of the diluted (10^7^ CFU/mL) secondary cultures of the different strains. The cultures were incubated continuously shaking for 24 h at 37 °C, with optical density (OD) being measured at 600 nm every 30 min in an OD reader (Synergy HT, BioTek Instruments, GmbH, Switzerland). Similar growth experiments were repeated using meat simulation media (MSM) instead of BHI. From the OD_600_ growth data, growth kinetics such as lag phase duration, maximal growth rate, and area under the curve (AUC) were determined using the program DMFit [88] and Graphad Prism, respectively. To assess for respiration as an indicator of growth or nutrient utilization the same experimental setup was also repeated but using BHI media supplement with 0.05% of the respiration indicator dye 2,3,5-triphenyltetrazolium chloride (TTC) (Sigma-Aldrich Chemie GmbH, Steinheim, Germany). To determine viable bacterial cell counts, cultures from each 96 well plate (from plates without TTC dye) were serially diluted in phosphate buffered saline (PBS), plated onto BHI plates, and incubated overnight at 37 °C. Viable cell counting was similarly performed on selected strains (covering the three genetic lineages and distributed over the entire observed PEA MIC range) that had been incubated in BHI at their respective PEA minimum inhibitory concentrations for 24 and 48 h. Viable bacterial cell counts were expressed in colony forming units (CFU) per ml of original broth (limit of detection: 1 CFU/mL; limit of quantitation: 25–250 CFU/plate). Normalized relative inhibition factors for each PEA concentration were calculated by dividing the viable bacterial cell counts (CFU/mL) of each strain by the counts of the un-supplemented (0 mg/mL PEA) positive controls. The mean and standard deviations across the biological replicates were calculated and plotted against the respective concentration. The above-mentioned PEA concentrations were also further accessed for their inhibitory power on solid media when formulated in BHI agar, this criterion yielded concentrations which were then used for the food matrix (lyoner and smoked salmon) assays. The water activity and pH of all working solutions containing PEA as well as the solvent were measured using an AQUALAB water activity meter (Decagon Devices Inc, Pullman, WA, USA) and pH-Indicator paper and pH-indicator strips (non-bleeding) (pH range 1 to 10) (Merck KGaA, Darmstadt, Germany). The pH values were also validated by measurement with a Thermo Scientific™ Orion™ Star A111 pH meter (Loughborough, UK).

#### Microscopy

The 100 µL aliquots of the cultures were spread onto microscopic slides, fixed and Gram-stained. Slides were microscopically examined for bacterial chaining levels with an Olympus BX40 microscope through a 100x/1.3 oil-immersion objective.

### 2.3. Effect of PEA on L. monocytogenes on RTE Food Models

Sliced lyoner sausage and Norwegian smoked salmon packages purchased from a local Swiss retailer were surface decontaminated using 70% ethanol. The packages were aseptically opened and the individual slices weighing 11 g each (lyoner slice diameter 10 cm) were placed on separate clean 145 × 20 mm petri dishes. A four-strain bacterial inoculum mixture (10^8^ CFU/mL) prepared in PBS was centrifuged (10 min at 6000× *g* and room temperature) and washed once in PBS. The inoculum strain selection included the *L. monocytogenes* EGDe reference strain, two isolates (N1546, N16-0044) from meat associated listeriosis outbreaks and strain LL195 from the 1983–1987 Swiss listeriosis outbreak. The concentration of the bacterial inoculum was confirmed by plating out serial dilutions onto BHI plates and determining the CFU/mL counts. To rule out possible pre-existing natural *L. monocytogenes* contamination in the lyoner and smoked salmon preparations controls of un-inoculated slices (one per experiment) were included within each experimental run. PEA weighing 82.5, 110, 165, and 220 mg were each dissolved in 250 μL sterile deionized water. Two hundred and fifty microliters of each PEA solution were applied in a dropwise manner over one side of the respective meat or salmon slices and evenly distributed using a sterile L-shaped spreader giving final PEA concentrations of 7.5, 10, 15, and 20 mg/g. As a positive control, one slice was similarly treated using sterile deionized water without PEA. The meat slices were dried at 4 °C. Then 11 μL of bacterial inoculum (10^8^ CFU/mL) was applied over the treated side of the meat slices and similarly spread. One uninoculated negative control, one deionized water treated inoculated positive control and PEA treated inoculated meat slices from each RTE meat type per experiment were stored for 7 and 11 days at 10 °C and 4 °C, respectively. Then the meat slices were separately placed in a sterile stomacher bag with 99 mL PBS and homogenized at normal grade for 2 min in a Laboratory Blender Stomacher 400 (Seward, Worthing, UK). Resulting homogenates were serially diluted and plated onto Oxoid chromogenic *Listeria* agar (OCLA) for *L. monocytogenes* enumeration. Plates were incubated at 37 °C and viable bacterial counts were determined after 24 h. To assess bacterial chaining under PEA stress 100 µL aliquots derived from the diluted homogenates were also microscopically examined. Viable bacteria cell counts determined from the PEA treated samples were divided by those determined for the non-PEA treated positive control to determine the relative inhibition factor. All experiments were repeated independently on three separate occasions. Results showing the mean and standard deviation from three biological replicates are presented.

### 2.4. Evaluation of Heat Stability of PEA Activity against L. monocytogenes

To assess the heat stability of PEA, lyoner production temperature conditions were simulated. MSM and BHI broth solutions containing 0 and 150 mg/mL PEA were heated to 80 °C in a water bath and held for 10 min at 80 °C. Thereafter, the broths were cooled and diluted to prepare working solutions supplemented with 0, 3.125, 6.25, 12.5, 16, 20, and 25 mg/mL PEA. These solutions were subsequently used in growth evaluation and MIC determination on a subset of 12 strains (Table 1) to determine PEA heat stability.

### 2.5. Biofilm Assay

The effect of PEA on biofilm formation was assessed at 25 °C in Tryptone soy broth (TSB) medium for a selection of 12 *L. monocytogenes* strains (Table 1). This strain selection represented all three *L. monocytogenes* lineages and covered the entire PEA MIC diversity of our strain collection and included the high biofilm producing *L. monocytogenes* strain N11-1850. Secondary TSB cultures were prepared from these strains as described above and diluted (1:40) in fresh TSB medium. A hundred microliters (10^7^ CFU/mL) of each of the diluted culture was added to respective wells of a flat bottomed 96-well polystyrene microtiter plate prefilled with an equal volume of TSB containing 0, 6.25 and 12.5 mg/mL PEA. Appropriate lids were placed on the plates and sealed with parafilm, and then biofilms were grown for 96 h at 25 °C, after which 100 µL from each well was transferred to a corresponding well of a new 96 well plate. The growth in each well was measured at 600 nm (using a Synergy HT OD reader) to allow for calculation of biofilm formation relative to growth. To remove unbound cells each well was rinsed 3 times with 200 µL sterile deionized water. At each rinse step, remnant wash solution was removed by inversion and gentle tapping of the 96 well plate on absorbent paper. The 96-well plates were dried for 30 min at 37 °C, after which cells bound in the biofilms were stained with 150 µL 1% aqueous crystal violet for 20 min. Excess stain was removed, and then the wells were rinsed 5 times with deionized water. After drying for 30 min at 37 °C, 100 µL of 96% ethanol was added to each well to dissolve the crystal violet within the bound stained cells. The dissolved stain was then quantified by measuring at 595 nm using an OD reader (Synergy HT, Biotek Instruments, GmbH, Switzerland).

### 2.6. Statistics

For all experiments done in this study, at least three independent biological replicates were performed unless stated otherwise. Statistical analysis of data was carried out using GraphPad Prism (Version 8.3.0 (328), GraphPad Software, San Diego, CA, USA). Significance of differences between the treatments or strains were identified through one-way ANOVA with post-hoc Tukey HSD tests, considering *p* values < 0.05 to be statistically significant.

## 3. Results

### 3.1. Evaluation of PEA Antimicrobial Effects against L. monocytogenes in BHI and MSM Broth Media

The antimicrobial effect of PEA was assessed in BHI broth revealing that it inhibited the growth of *L. monocytogenes* in a concentration and strain dependent manner. Significant growth inhibition was observed starting as low as 1.56 mg/mL PEA (Figure 1A). PEA prolonged lag phase and reduced growth rate and overall AUC (Appendix A). The MICs determined across the 62 *L. monocytogenes* strains examined ranged from 8 to 12.5 mg/mL (Figure 1B, Table 1 and Table 2; Appendix A). Examining PEA inhibition on BHI agar also showed similar MICs and trends for the strains as observed in BHI broth (Table 1; Appendix A). PEA supplementation of the BHI and MSM working solutions did not alter their overall pH and water activity (Appendix A). PEA inhibition was also evaluated in MSM broth showing that although maximum cell densities achieved by the *L. monocytogenes* strains were lower in this media than in BHI the PEA MICs observed for the strains were similar (Table 1). While assessing growth in TTC supplemented BHI broth there was no respiratory activity detected in *L. monocytogenes* cultures that were cultivated at PEA concentrations that were equivalent or above the strain specific MIC, indicating that the PEA applied at such concentrations did not only inhibit growth but also bacterial respiration or metabolic activity (data not shown). Determination and comparison of viable bacterial cell counts before and after exposure to PEA at MIC showed that in addition to growth inhibition PEA also had a bactericidal effect on *L. monocytogenes* (Figure 2; Appendix A). The observed bactericidal effects varied between the strains ranging from 1 to 8 log CFU/mL for a tested selection of 11 strains observed at 24 and 48 h of incubation at strain specific PEA MICs in BHI broth (Figure 2). Notably, the strain *L. monocytogenes* EGDe used as a reference strain turned out to be the most sensitive exhibiting the highest sensitivity to PEA bactericidal effects and having the lowest PEA MIC amongst the tested strains (Table 1; Figure 2). When assessing the stability of PEA inhibitory effects under simulated lyoner heat treatment conditions in both MSM and BHI broths we observed that these heat treatment conditions (10 min at 80 °C) had no negative impact on the PEA inhibitory capacity as similar MICs were determined for selected strains before and after the heat treatment (Table 1; Appendix A). Overall there were no observable differences in bacterial chaining observed among the strains for all the different experimental setups (data not shown), ruling out the possibility that differences observed in the viable bacterial cell (CFU) counts in our study could have been distorted due to differences in bacterial chaining under the different PEA treatment conditions.

### 3.2. PEA Is an Effective Inhibitor of L. monocytogenes on Meat and Salmon

The antimicrobial activity of PEA against *L. monocytogenes* on RTE foods was assessed using lyoner and cold smoked salmon products as models. Lyoner sausage and smoked salmon slices surfaced coated with varying PEA concentrations were inoculated with *L. monocytogenes* (1.56 × 10^5^ CFU/slice giving approximately 1.36 × 10^4^ CFU/g) and stored at 4 °C and 10 °C for 11 and 7 days, respectively. No *L. monocytogenes* was detected on all the negative control slices of the lyoner sausage and salmon that had not been PEA treated and not inoculated, indicating that all meat batches used in our experiments, if any, had levels of natural *Listeria* contamination that were below the detection limit of the assay used. On positive control meat samples inoculated with *Listeria* but not PEA treated the *L. monocytogenes* inoculum had grown by an average of 2.58 log CFU on lyoner slices, but no growth was observed on the smoked salmon slices after 11 days of storage at 4 °C (Figure 3A,B and Figure 4A,B; Appendix A). Whereas, in those samples incubated 7 days at 10 °C the positive control samples had average growths of 3.55 and 0.21 log per slice on the lyoner and smoked salmon samples, respectively (Figure 3C,D and Figure 4C,D; Appendix A). As such, *L. monocytogenes* grew better on lyoner than on salmon and growth observed at 10 °C was greater than that at 4 °C in the absence of PEA treatment (Figure 3 and Figure 4). PEA application caused *L. monocytogenes* growth inhibition on both lyoner and salmon slices (Figure 3 and Figure 4; Appendix A). Compared to the positive control samples there was significant *L. monocytogenes* growth inhibition observed on the lyoner slices incubated at both 4 °C and 10 °C (Figure 3). Meanwhile in the case of PEA treated salmon slices statistically significant inhibition of *L. monocytogenes* growth was observed in samples incubated at 10 °C whilst at 4 °C significant difference was only noted on the highest tested PEA concentration (20 mg/g) in comparison to the positive control (Figure 4).

The PEA treatment caused a 0 to 2.12 log_10_ CFU and 0 to 2.1 log_10_ CFU reduction in final bacterial cell counts at concentrations between 7.5 mg/g to 20 mg/g on lyoner at 4 °C and 10 °C, respectively, in comparison to the cell counts from the positive control (Figure 3). Meanwhile, on smoked salmon PEA treatment caused a 0 to 0.17 log_10_ CFU and 0 to 0.35 log_10_ CFU reduction in final bacterial cell counts at concentrations between 7.5 mg/g to 20 mg/g at 4 °C and 10 °C, respectively, in comparison to the cell counts from the positive control (Figure 4). On smoked salmon the pieces had lower cell counts than what they had been inoculated with, indicating that PEA in addition to inhibiting cell division, also impacted the survival of the bacteria (Figure 4; Appendix A). This reduction trend increased with increasing PEA concentration but was only statistically significant at 20 mg/g. At the lowest concentration (7.5 mg/g) tested, PEA had no longer an effect on bacterial cell count, in fact, it appeared the counts were higher than those of the positive control though the difference was not statistically significant (Figure 3 and Figure 4; Appendix A). Overall there were no observable differences in bacterial chaining observed among the strains for all the different meat and salmon experimental setups, ruling out the possibility that differences observed in CFU counts could have been distorted due to differences in bacterial chaining under the different PEA treatment conditions.

### 3.3. PEA Reduces L. monocytogenes Biofilm Production

Since biofilm production can facilitate the long-term persistence and survival of isolates in the food processing plants the effect of PEA on biofilm production was assessed. PEA applied at 3.13 and 6.25 mg/mL caused a significant reduction of biofilm production in a strain dependent manner (Figure 5; Appendix A). Notably, the greatest PEA inhibitory effect on biofilm production was observed on the high biofilm producing strains (N11-1850, N12-1772 and N13-0119 (Figure 5). Biofilm forming ability was variable amongst the tested strains and inhibition thereof by PEA did not follow any particular genetic background associated trend.

## 4. Discussion

In this study, we determined that PEA was effective at inhibiting growth and reducing viable cell counts of *L. monocytogenes* on BHI agar and in BHI broth as well as in MSM broth. The anti-listeria activity of PEA was not affected by pasteurization conditions, which is key to its potential application in the recipe for heat treated products. PEA also decreased the capacity of *L. monocytogenes* to produce biofilms. The results, moreover, highlight the need to incorporate a good selection of strain diversity for the development and validation of interventions for *L. monocytogenes* control on foods. As shown here the widely used *L. monocytogenes* EGDe reference strain had the highest PEA sensitivity amongst the tested strains. In comparison to commonly used food additives with antimicrobial activity such as nisin and sodium chloride which mainly have bacteriostatic activity, PEA has both bacteriostatic and bactericidal activity though significantly higher concentrations are required for similar bacteriostatic activity in comparison to nisin [47,48].

Results from in vitro studies are, however, limited to mimic the situation in different food products. Therefore, we performed an additional study trial using selected foods. These experiments on a bologna type sausage (lyoner) and cold smoked salmon pieces showed that PEA had inhibitory and bactericidal activity against *L. monocytogenes* on these products. The effect was more pronounced in the liquid media than on the food products signifying the importance of confirming in vitro findings on the end products intended for the intervention. However, the trends demonstrated both, in liquid media and the food products, indicated increased inhibition with increasing PEA concentration. As previously described for other antimicrobials, the reduced activity on food products vs. liquid media or agar might be influenced by factors such as uneven distribution or low solubility in the food matrix, adsorption of the antimicrobial to food constituents, and inactivation or degradation by enzymes or commensal microbiota [38,89].

*L. monocytogenes* seems to be more sensitive to PEA stress on food matrices in comparison to some microorganisms as significant PEA bacteriostatic or bactericidal activity was observed at lower concentrations (10 to 20 mg/g PEA) whilst for *E. coli* it was observed at higher levels (≥70 mg/mL) [65,69].

The lack of a stronger inhibitory effect of PEA on *L. monocytogenes* on salmon than on lyoner could be linked to uneven distribution of PEA on the food matrix. Future analysis might need to test different PEA application methods such as injection during processing or fine mist spray to improve the distribution of PEA. Furthermore, as alluded to above some cold smoked salmon matrix intrinsic factors including commensal microbiota could have contributed to the decreased PEA antimicrobial activity observed on salmon. More work needs to be done to determine how further efficiency can be achieved by using higher concentrations of PEA or when combining this intervention with other hurdle techniques such as salt, nisin, high-pressure treatment, and vacuum packaging.

The contamination levels of *L. monocytogenes* used in our study are significantly higher than usual since initial natural contamination levels when present are usually below 10 CFU/g [23]. The higher contamination levels applied in this study were chosen to enable the detection and quantification of the effects of the interventions on the bacteria. Furthermore, we wanted to limit the “Jameson effect” due to the competitive microbiota of the foods as microbial growth might be greatly inhibited once the predominant bacteria species has reached its maximum population density [90].

The formation of nonculturable but viable *L. monocytogenes* cells during the PEA application could be an explanation for what appears as a bactericidal effect observed on salmon stored at 4 °C. However, in the BHI growth assays, we observed an absence of respiration or metabolic activity using TTC dye, hence it can be assumed that the bacteria had died. Moreover, stressors such as low temperature, pH and osmotic stress can induce chaining or filamentation in *L. monocytogenes* due to incomplete cell separation post cell division [91,92]. This would lead to an underestimation of the actual bacteria numbers in direct plate counts because several cells in a chain will produce a single colony. Hence, the probability of chaining must be taken into account when drawing conclusions from reduced cell counts of *L. monocytogenes* on foods after adding a stressor. In our study, however, the analysis for chaining revealed that PEA exposure did not result in any observable increase in chaining as determined by analysis of Gram stained slides.

The effects of most salts such as sodium lactate have been ascribed to pH and water activity reduction [93]. In our study, no significant changes in pH or water activity in the PEA treated samples were observed. The values where at a level at which no impact on *L. monocytogenes* growth would be expected. It seems PEA exerts an antimicrobial effect on the growth of *L. monocytogenes* that is independent of changes in pH and water activity.

Attempts to influence bacterial signal transduction pathways with substances such as PEA have been investigated in a selection of pathogens. For example, some infections by *E. coli* can be treated through quorum sensing and two-component signaling systems targeting antimicrobials such as LED209 and ZFH-02056 [94,95,96,97]. It has also been suggested by others that a similar system could be responsible for *E. coli* response to PEA stress [65,68]. This two-component system might transmit signals to multiplication and cell division probably mimicking the signals of a culture in stationary phase thereby inhibiting multiplication and possibly inducing autolysis. Going forward our research will focus on detailing the mechanism of PEA action.

High biofilm production ability and resistance to disinfectants are key for *L. monocytogenes* persistence in food processing environment [3,54,59,63,64]. Biofilms offer protection from the harsh environment that exists in the food processing environment. The presence of high biofilm producing *L. monocytogenes* strains, coupled with unhygienic substandard processing practices and poor maintenance and sanitation of processing plants, are major contributors to contamination of foods with *L. monocytogenes* [3,15,37]. From our data, *L. monocytogenes* biofilm production can be reduced by PEA, hence integrating PEA into cleaning routines especially on food contact surfaces where some disinfectants might not be suitable could contribute to reduction of contamination levels and persistence of *L. monocytogenes* in food processing plants. Although not all strains biofilm ability was reduced, the concentrations used were below the MIC and some of these strains might have required a higher concentration for biofilm production to be inhibited, future studies need to clarify this. However, a positive aspect is that the high biofilm formers were significantly inhibited in their biofilm forming ability at low concentrations. The biofilm experiments were done in vitro using attachment surfaces that might be different from those existing within some parts of the processing plant environment, hence any conclusions must be made with this limitation in mind.

The potential application of PEA might be exploited as a safe food bio preservative for use with substances such as nisin to which resistance is rising thereby minimizing the *L. monocytogenes* nisin-resistance problem [43,44,45,47]. As previously suggested by others, one possible way to use PEA is to add it during product formulation or injecting or spraying the food product as a decontamination treatment [65]. Another attractive application of PEA would be to put high concentrations of PEA in packaging films and protective coatings as was shown with sodium lactate (NaL) plus nisin, NaL plus sodium diacetate, and mustard extract or sinigrin impregnated films that prevented *L. monocytogenes* growth on cold smoked salmon and bologna sausages, respectively [98,99,100]. This technique becomes more relevant were antimicrobial ingredient addition to the food product during formulation is not desirable. Another alternative could be to use it to treat food contact surfaces to avoid cross-contamination of products. Since food contact surfaces provide an ideal environment for pathogenic bacteria such as *L. monocytogenes* to form biofilm on [101,102,103], PEA can also be used to treat these as we could show that it reduces *L. monocytogenes* biofilm production. Future studies must tackle the practical nature of specific applications, treatments, and combinations which would be synergistic or additive minimizing induction of cross protection such as is observed with NaCl and nisin at low temperatures [43,45]. Since the effects of PEA appear to be concentration dependent, maximum legally permissible concentrations that do not alter product quality must be established for each product. Likewise, the effectiveness against other food contaminating pathogens beyond *L. monocytogenes* in the context of RTE foods must be determined.

Potential drawbacks to application of amines such as PEA in foods include their potential negative health effects to segments of the population with limited amine degradation capacity due to factors that are hereditary or associated with alcoholism and use of monoamine oxidase inhibiting medications [104]. Future research must address these questions especially the determination of maximum allowable inclusion levels that avoid these negative biological effects. In food products that undergo high heat or smoke treatment and are exposed to acidic conditions the potential of carcinogenic nitrosamine formation in meat products containing nitrates or nitrites might be a limiting factor to PEA application in such products [105]. However, to circumvent this problem, PEA could potentially be applied post heat or smoke treatment or via other alternative application methods as described above in such products. It is important to note that the final concentration, if any, of nitrosamines that might be formed would be influenced by heating or cooking method, pH, and duration of the process applied [105]. Limiting cooking time and addition of natural nitrosation inhibiting substances such as the vitamins, alpha-tocopherol, and ascorbic acid might, therefore, be applied to mitigate this risk [105]. It remains to be determined whether PEA increases or induces nitrosamine formation in those products where a risk of their formation could be introduced or increased by PEA addition.

In conclusion, our data suggest that PEA could be applied as an anti-listerial natural food additive. Based on good manufacturing practices as a prerequisite, the application of PEA to RTE products might be an additional hurdle to limit *L. monocytogenes* growth, thereby increasing food safety.

## Figures and Tables

**Figure 1 foods-09-01363-f001:**
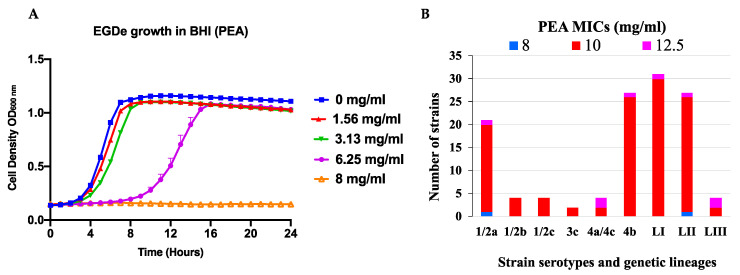
Evaluation of β-phenylethylamine (PEA) inhibitory effect against *L. monocytogenes*. (**A**) Growth curves of *L. monocytogenes* EGDe in brain heart infusion (BHI) broth supplemented with increasing PEA concentrations at 37 °C. Presented are growth curves from a representative reference strain EGDe plotted from mean OD_600_ measurement from three replicate experiments observed from kinetic growth assays in BHI supplemented with PEA. (**B**) Distribution of PEA minimum inhibitory concentrations with respect to serotypes and genetic lineages (L) of the 62 *L. monocytogenes* strains examined.

**Figure 2 foods-09-01363-f002:**
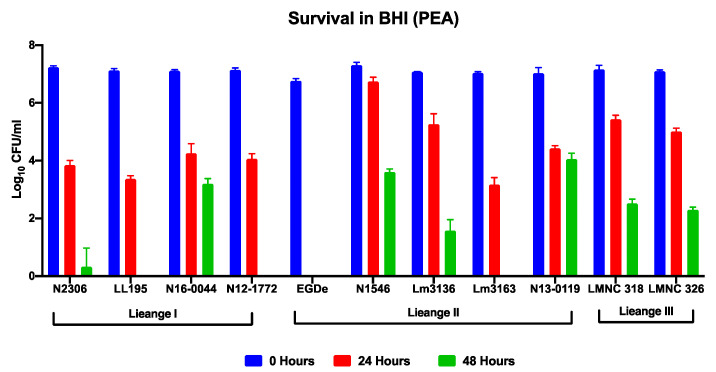
PEA has bactericidal activity against *L. monocytogenes*. Survival rates of a representative selection of 11 *L. monocytogenes* strains determined after 24 and 48 h of exposure to PEA MIC in BHI at 37 °C. Presented data show mean log_10_ colony forming units (CFU) counts (bars) and standard deviation of three biological replicates at 0 (start of incubation), 24 and 48 h of incubation at PEA MIC. For all strains, there was a significant difference between cell counts at each of their respective sampling time points at *p* < 0.05 according to one-way ANOVA and Tukey post-hoc test pairwise comparison of all the treatment conditions.

**Figure 3 foods-09-01363-f003:**
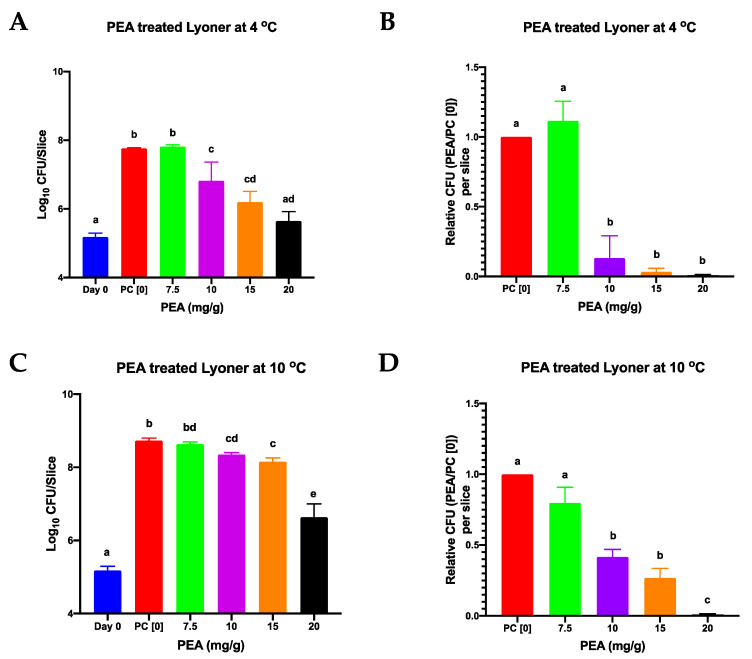
Effect of PEA on bacterial counts on lyoner sausage treated with different PEA concentrations and contaminated with a cocktail of four *L. monocytogenes* strains and stored for 11 and 7 days at 4 °C (**A**,**B**) or 10 °C (**C**,**D**), respectively. (**A**,**C**): Viable bacterial counts were determined and expressed in Log CFU per slice. (**B**,**D**): CFU levels determined per PEA lyoner slice expressed relative to those of the positive control with no PEA added (PC [0]) slices. PC [0] denotes the positive control sample (PEA untreated, *L. monocytogenes* inoculated slices). Data shown are the mean (bars) and standard deviation of three biological replicates. Different letters indicate significant differences between treatments that were identified through one-way ANOVA and Tukey post-hoc test pairwise comparison of all the treatment conditions (*p* < 0.05).

**Figure 4 foods-09-01363-f004:**
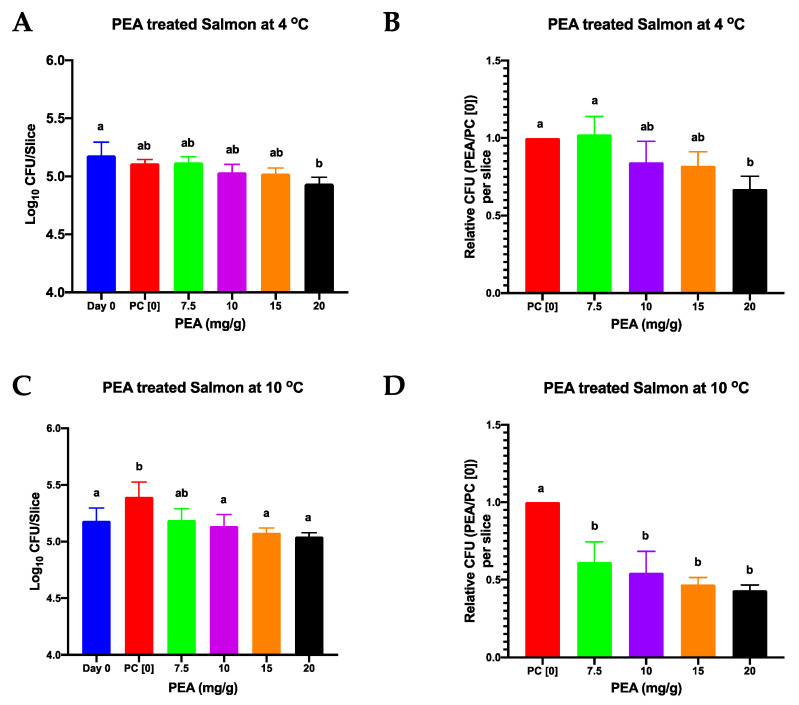
Effect of PEA on bacterial counts on smoked salmon slices treated with varying PEA concentrations and contaminated with a cocktail of four *L. monocytogenes* strains and stored for 11 days at 4 °C (**A**,**B**) or 7 days at 10 °C (**C**,**D**). (**A**,**C**): Viable bacterial counts were determined and expressed in Log CFU per slice. (**B**,**D**): CFU levels determined per PEA smoked salmon slice expressed relative to those of the PC [0] slices. PC [0] denotes the positive control sample (PEA untreated, *L. monocytogenes* inoculated slices). Data shown are the mean (bars) and standard deviation of three biological replicates. Different letters indicate significant differences (*p* < 0.05) that were identified through one-way ANOVA and Tukey post-hoc pairwise comparison test of all the treatment conditions.

**Figure 5 foods-09-01363-f005:**
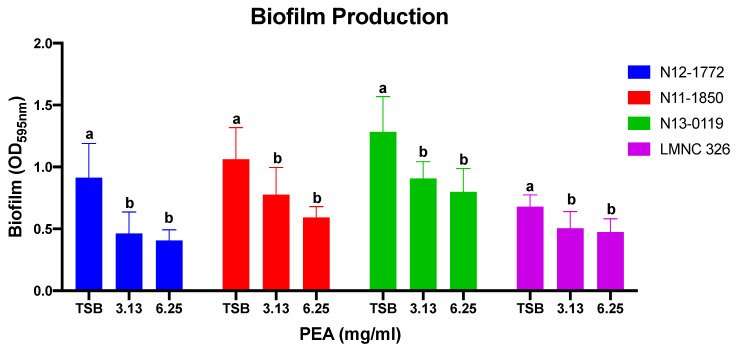
PEA reduces biofilm production. Presented data show the mean (bars) and standard deviation (error bars) of three independent biological experiments for selected strains representing lineage I, II and III. Data for biofilm formation were normalized relative to growth (OD_600nm_ reading at 96 h). For each strain different letters indicate significant difference between biofilm produced when grown in TSB compared to when grown in TSB supplemented with PEA at sub-MIC concentrations. *p* < 0.05 based on one-way ANOVA and Tukey post-hoc test pairwise comparison of all the strains.

**Table 1 foods-09-01363-t001:** Study strains and their PEA MIC in different media.

Strain ID	Source	ST ^d^	CC ^d^	Serotypes	Lineage	Reference	PEA MIC (mg/mL) ^c^
N2306 ^**a**,**b**^	Ready-to-eat salads	ST4	CC4	4b	I	[9]	10
LL195 ^**a**,**b**^	Vacherin Montd’or cheese	ST1	CC1	4b	I	[79]	10
N14-195	Meat/Meat product	ST31	CC31	4b	I	[80]	10
N16-0044 ^**a**,**b**^	Meat pâté	ST6	CC6	4b	I	[11]	10
H34	Human listeriosis	ST489	CC489	1/2b	I	[81]	10
N11-2292	human listeriosis	ST1	CC1	4b	I	[82]	12.5
N11-2675	human listeriosis	ST1063	CC5	1/2b	I	[82]	10
N14-0435	Milk product	ST3	CC3	1/2b	I	[80]	10
N14-0487	Plant associated	ST4	CC4	4b	I	[80]	10
N12-0605	Meat/Meat product	ST727	CC1	4b	I	[80]	10
N12-1339	Meat/Meat product	ST746	CC1	4b	I	[80]	10
N12-1996	Milk/Milk product	ST1	CC1	4b	I	[80]	10
N13-0047	Milk/Milk product	ST1	CC1	4b	I	[80]	10
Scott A	Human listeriosis	ST290	CC2	4b	I	[83]	10
N11-2747	human listeriosis	ST1	CC1	4b	I	[82]	10
N12-0341	human listeriosis	ST1	CC1	4b	I	[82]	10
N12-0551	human listeriosis	ST1	CC1	4b	I	[82]	10
N12-0320	human listeriosis	ST4	CC4	4b	I	[82]	10
N12-0794	human listeriosis	ST4	CC4	4b	I	[82]	10
N13-2107	Meat/Meat product	ST4	CC4	4b	I	[80]	10
N13-1054	human listeriosis	ST1285	CC2	4b	I	[82]	10
N12-1387	human listeriosis	ST6	CC6	4b	I	[82]	10
N11-2801	human listeriosis	ST6	CC6	4b	I	[82]	10
N12-1772 ^**a**^	Milk/Milk product	ST682	CC4	4b	I	[80]	10
N13-1184	Meat/Meat product	ST6	CC6	4b	I	[80]	10
N12-0973	Meat/Meat product	ST2	CC2	4b	I	[80]	10
N11-1846	Meat/Meat product	ST724	CC2	4b	I	[80]	10
N12-0432	Meat/Meat product	ST2	CC2	4b	I	[80]	10
N11-1850 ^**a**^	Milk/Milk product	ST1290	CC217	4b	I	[80]	10
N12-0466	Meat/Meat product	ST2	CC2	4b	I	[80]	10
N12-1608	human listeriosis	ST224	CC224	1/2b	I	[82]	10
Lm1043S	Human listeriosis	ST85	CC7	1/2a	II	[84]	10
N1546 ^**a**^	Imported cooked ham	ST8	CC8	1/2a	II	[8]	10
Lm3136 ^**a**^	Tomme cheese	ST18	CC18	1/2a	II	[85]	10
Lm3163 ^**a**^	Tomme cheese	ST26	CC26	1/2a	II	[85]	10
N586	Human prosthetic joint	ST412	CC412	1/2a	II	[74]	10
N843	Human prosthetic joint	ST412	CC412	1/2a	II	[74]	10
EGDe ^**a**,**b**^	Rabbits	ST35	CC9	1/2a	II	[86]	8
N11-1515	Milk product	ST29	CC29	1/2a	II	[80]	10
N11-1617	Meat/Meat product	ST8	CC8	1/2a	II	[80]	10
N11-2183	Plant associated	ST20	CC20	1/2a	II	[80]	10
D:824/5	Meat product	ST9	CC9	3c	II	[80]	10
N11-1514	Meat/Meat product	ST9	CC9	1/2c	II	[80]	10
N12-1921	Plant associated	ST9	CC9	1/2c	II	[80]	10
D: 650/8	Meat/Meat product	ST9	CC9	3c	II	[80]	10
N12-0152	Milk/Milk product	ST9	CC9	1/2a	II	[80]	10
N11-1837	human listeriosis	ST9	CC9	1/2a	II	[82]	10
N12-0486	human listeriosis	ST9	CC9	1/2c	II	[82]	10
N13-0001	human listeriosis	ST9	CC9	1/2c	II	[82]	10
N12-1864	Milk/Milk product	ST9	CC9	1/2a	II	[80]	10
N11-1649	Meat/Meat product	ST743	CC8	1/2a	II	[80]	10
N11-1584	human listeriosis	ST1295	CC8	1/2a	II	[82]	10
N12-1273	Human listeriosis	ST412	CC412	1/2a	II	[82]	10
N11-1346	Human listeriosis	ST673	CC8	1/2a	II	[82]	12.5
N11-1905	Meat/Meat product	ST121	CC121	1/2a	II	[80]	10
N12-1024	Meat/Meat product	ST121	CC121	1/2a	II	[80]	10
N13-0119 ^**a**^	human listeriosis	ST121	CC121	1/2a	II	[82]	10
N12-0367	human listeriosis	ST121	CC121	1/2a	II	[82]	10
WSLC1019	Animal isolate	ST130	CC69	4c	III	ATCC 19116	10
LMNC318 ^**a**^	Ruminant listeriosis	ST70	CC70	4a/4c	III	[87]	12.5
LMNC326 ^**a**^	Ruminant listeriosis	ST70	CC70	4a/4c	III	[87]	12.5
WLSC1020	Animal isolate	ST71	CC131	4a	III	ATCC 19114	10

^**a**^ Strains used in the biofilm assays as well as assays to evaluate heat stability of PEA activity against *L. monocytogenes*. **^b^** Strains used in assays to evaluate the effect of PEA on *L. monocytogenes* using food models. **^c^** PEA MIC for each strain in BHI broth, BHI agar, meat simulation media (MSM), heat treated BHI (HT BHI) and heat treated MSM (HT MSM). **^d^** ST: sequence type, CC: clonal complex.

**Table 2 foods-09-01363-t002:** Serotype, lineage, and PEA MICs.

	PEA MIC ^a^			Total
	8 mg/mL	10 mg/mL	12.5 mg/mL	
Serotype				
1/2a	1	19	1	21
1/2b	-	4	-	4
1/2c	-	4	-	4
3c	-	2	-	2
4a/4c	-	2	2	4
4b	-	26	1	27
Lineage				
LI	-	30	1	31
LII	1	25	1	27
LII	-	2	2	4

^**a**^ PEA MIC reported were determined in BHI broth, BHI agar, and MSM.

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
