# Peer review of "β-Phenylethylamine as a Natural Food Additive Shows Antimicrobial Activity against Listeria monocytogenes on Ready-to-Eat Foods"

_foods, 2020, doi:10.3390/foods9101363_

Round 1
Reviewer 1 Report
This is a study on the antimicrobial effect of β-phenylethylamine (PEA) on Listeria monocytogenes with the aim to use this compound for food decontamination. By using a significant number of Listeria strains, the authors have determined the minimum inhibitory concentration and the effect on biofilm formation of PEA in vitro. Antimicrobial effect has also been assessed on two food products, a Bologna type product (lyoner) and smoked salmon, as well as the influence of heat treatment on PEA effect.
General comment
The growing interest in finding new/alternative antimicrobials for the food industry has boosted extensive research on this area, and many compounds are under study. However, the use of amines such as PEA raises some questions about their potential use as antimicrobials in foods. PEA has been tested for some applications, such as disinfection of medical devices, and some preliminary experiences have been performed in foods. However, biogenic amines creates great concern in foods due to their biological effects. Furthermore, in products containing or added with e.g. nitrite (as it is the case of cured meat products), N-nitroso compounds such as nitrosamines can be formed, even more if products are heated. Even if PEA has a GRAS status, it is not accepted as food additive in the European Union. Re-evaluation of food additives is also continuously underway. The authors should address these considerations in the introduction and/or in the discussion of the manuscript.
Specific comments
Title
There is a mistake in the name of the microorganism in the title: “monocytogens”.
Methods
What was the limit of detection of the enumeration technique of viable bacterial cell counts?
Please, specify in the text the criteria for the selection of strains in the different studies.
L133-135: pH would not be expected to dramatically change after PEA addition, but the use of a pH-meter would be necessary to obtain accurate measurements. Alternatively, more information about the measurement range of the indicator paper/strips should be provided.
Results
L258: Please, revise the sentence. Listeria was not detected, but this does not necessarily mean that samples were “free of natural Listeria contamination”. As above mentioned, please include information about the limit of detection of the enumeration technique in M&M.
Discussion
Some parts of the discussion should be improved, i.e, by including comparison of the inhibition achieved with other antimicrobials, the use of PEA by other authors, the possible biological effects of PEA, etc.
L345-348: Other hypotheses could be explored: any effect of smoking?
L349-350: Please, update the reference. There are more recent studies about natural contamination levels of Listeria in foods.
L367-370: aw was not determined. Furthermore, discussion about the possible inhibitory mechanism of PEA from the literature could be provided.
L412-415: Further research is needed to reach this conclusion.
Figures and tables
Please, use letters in all the figures to show significant differences between batches.
For a better comprehension, in Figures 3 and 4 “Inoculum” could be replaced by “Day of inoculation” or “Day 0”.
References
Some parts of the manuscript contain an excessive number of citations (e.g. L44, L52, L61, etc.), which also results in an extensive reference list for a research article. The authors should select the most relevant and appropriate citations. Also, in some cases the authors should better refer to the original source, e.g. in L38-40, reference 23 (in these case, a reference to EU regulation would be more appropriate).
Other comments
Replace “flora” and “microflora” by “microbiota” throughout the text.
Reviewer 2 Report
The manuscript addresses a topic worthy of investigation. In detail, the paper is interesting and give additional data about the antimicrobial activity against Listeria monocytogenes on ready-to-eat-foods. I have only some minor suggestions to improve the quality of the paper.
The introduction offers a good overview of the state of art. However, the introduction should be implemented taking into account Listeria stress resistance and in particular the universal stress proteins that allow the growth even in acid conditions of many RTE foods.
The methodology is sound and the results are of concern. Discussion of the results is well reported and reasoned.
Reviewer 3 Report
The objective of the study is to evaluate the inhibitory effect of β-phenylethylamine against Listeria monocytogenes in model systems and in ready to eat food products. The study is interesting and well designed. The manuscript is well written and data analysis is adequate and supported by appropriate statistical analysis. I recommend acceptance of the article after a careful revision of English language throughout the manuscript.
Round 2
Reviewer 1 Report
The authors have improved the manuscript after the revision, but I enclose a few more comments prior to acceptance.
L19-20: The sentence “In conclusion, our data confirmed PEA as an anti-listerial natural food additive” should be deleted from the abstract. Much more research is needed to reach this conclusion, and the text has been revised in this sense.
L507-508: Please, revise: “In conclusion, our data suggest that PEA could be applied as an anti-listerial natural food additive.”
L297: Replace “free of” by “below”.
